# Microstructure Evolution of the Ti-46Al-8Nb-2.5V Alloy during Hot Compression and Subsequent Annealing at 900 °C

**DOI:** 10.3390/ma16186176

**Published:** 2023-09-12

**Authors:** Shouzhen Cao, Zongze Li, Jiafei Pu, Jianchao Han, Qi Dong, Mingdong Zhu

**Affiliations:** 1School of Electrical and Mechanical Engineering, Huangshan University, Huangshan 245021, China; caoshouzhende@163.com (S.C.); pujiafei91@163.com (J.P.); dq88637@163.com (Q.D.); 2Engineering Research Center of Advanced Metal Composites Forming Technology and Equipment of Ministry of Education, Taiyuan University of Technology, Taiyuan 030024, China; lee980403@163.com; 3Science and Technology on Reactor System Design Technology Laboratory, Nuclear Power Institute of China, Chengdu 610213, China; zhumingdong2008@163.com

**Keywords:** TiAl alloy, hot compression, service temperature annealing, microstructure evolution, phase transformation

## Abstract

TiAl alloys are high-temperature structural materials with excellent comprehensive properties, and their ideal service temperature range is about 700–950 °C. High-Nb containing the Ti-46Al-8Nb-2.5V alloy was subjected to hot compression and subsequent annealing at 900 °C. During hot compression, work-hardening and strain-softening occurred. The peak stresses during compression are positively correlated with the compressive strain rates and negatively correlated with the compression temperatures. The α_2_ phase exhibited a typical (0001)α_2_ basal plane texture after hot compression, while the β_0_ and γ phases did not show a typical strong texture. Subsequent annealing at 900 °C of the hot-compressed samples resulted in significant phase transformations, specifically the α_2_ → γ and β_0_ → γ phase transformations. After 30 min of annealing, the volume fraction of the α_2_ phase decreased from 39.0% to 4.6%. The microstructure characteristics and phase fraction after 60 min of annealing were similar to those after 30 min. According to the calculation of Miller indexes and texture evolution during annealing, the α_2_ → γ phase transformation did not follow the Blackburn orientation relationship. Multiple crystal-oriented α_2_ phases with nanoscale widths (20~100 nm) precipitate within the γ phase during the annealing process, which means the occurrence of γ → α_2_ phase transformation. Still, the γ → α_2_ phase transformation follows the Blackburn orientation relationship.

## 1. Introduction

Intermetallic compounds have a wide range of applications in aerospace as high-temperature structural materials. The TiAl alloy has attracted considerable attention from material researchers for its low density (3.3–4.2/cm^3^), high strength, excellent high-temperature oxidation resistance, creep resistance, and flame retardant properties, and is considered an ideal structural material to replace nickel-based superalloys at 700–950 °C [1,2,3,4,5]. The main phases of the TiAl alloy are the α_2_-Ti3Al phase (D0_19_, P63/mmc, a = b = 5.793, c = 4.649), the γ-TiAl phase (L1_0_, P4/mmm, a = b = 3.976, c = 4.049), and the β_0_-Ti phase (B2, Im-3m, a = b = c = 3.306) [6]. The lattice parameters and composition vary among different phases. Significant phase transformation occurs when the TiAl alloy is heat treated above the eutectoid transition temperature (T_eu_, between about 1100 °C and 1200 °C) [1,7,8,9]. The phase transformation rate and trend are primarily influenced by temperature conditions, a non-equilibrium state induced by the cooling rate, and external stress conditions. The service temperature of the TiAl alloy is 700–950 °C, considerably lower than T_eu_. The primary service environment for the TiAl alloy is high-temperature heat exposure or thermal stress conditions [10,11].

The phase diagram is an essential guide for studying phase transformation, microstructure, and mechanical properties of the TiAl alloy. The decomposition of the α_2_ phase is the primary characteristic of phase transformation at the service temperature [12,13,14]. However, the properties of the α_2_ phase decomposition vary significantly under different driving conditions. Huang et al. conducted extensive experiments on the α_2_ phase at the service temperature, subjecting it to thousands of hours of thermal exposure. Their findings revealed that the α_2_ phase within the lamellar colonies exhibited an intermittent shape and underwent a narrow range of the α_2_ → γ phase transformation [15,16,17,18]. The higher the degree of deviation of elements in a phase, the more intense the phase transformation at service temperature. Under the substantial element deviation, the TiAl alloy will undergo significant phase transformation quickly. The phase diagram shows the phase transformation in an equilibrium state, and the phase transformation temperature in a non-equilibrium state is often lower than T_eu_ [19]. The Ti-45Al-8.5Nb-0.2W-0.2B-0.02Y alloy was quenched at 1400 °C and obtained a microstructure containing many non-equilibrium phases. After annealing for 800 °C/20 min, apparent phase transformation occurred, and the ω0 phase was uniformly precipitated near the nanoscale (α_2_ + γ) lamella. This phenomenon often takes hundreds of hours in the microstructure of heat exposure [20]. The complete α phase massive transformation was observed in the rapidly cooled Ti-47Al-2Cr-2Nb powder. Apparent phase transformation then occurred after being held at 700 °C/900 °C for 30 min, forming nanoscale plate phases and proving the existence of the 6H type Ti_2_Al phase and its transformation relationship with α_2_ and γ phases [21].

Stress is a meaningful way to drive the recrystallization phase transformation of the TiAl alloy, especially since the thermal–mechanical process of the TiAl alloy will undergo significant phase transformation [22,23]. Above the T_eu_, TiAl alloys have excellent hot working properties, and the stress of large plastic deformation can induce significant dynamic recrystallization and microstructure refinement [24,25,26]. In fatigue and creep experiments of the TiAl alloys below the T_eu_, the pressure can significantly accelerate the service decomposition of the α_2_ phase. The α_2_ → γ phase transformation occurs in traditional the TiAl alloys in different service conditions, but the morphological and orientation relationship characteristics of these phase transformations are considerably different in different modes [27,28,29,30]. There are few studies on the microstructure instability and phase transformation of the TiAl alloy with high-stress stored energy at service temperature. This study aims to investigate the strain-induced phase transformation and microstructure evolution of TiAl alloys to improve the understanding of the service performance of TiAl alloys under complex service conditions.

## 2. Materials and Methods

The Ti-46Al-8Nb-2.5V alloy was smelted by vacuum induction in a water-cooled copper crucible. The raw materials are high-purity sponge titanium, high-purity aluminum, the Al-Nb intermediate alloy (Nb52.4 wt%), and the Al-V intermediate alloy (V 53.88 wt%). Considering the influence of Al element volatilization loss and water-cooled condensation shell, the raw material formula is modified and put into the copper crucible. The melting process is carried out under a vacuum atmosphere of less than 2.0 Pascal. The melting power used in the smelting process was slowly increased to 500 KW. After the raw material was completely melted, the holding time lasted for 10 min, and then the melt was cast into the investment casting shell of the ZrO_2_ surface layer. The shell was preheated at 800 °C to obtain the TiAl alloy casting. After a series of shell breaking, cleaning, and sandblasting, the TiAl alloy castings were obtained. A sample of Φ8 × 12 mm was prepared by wire cutting, polished, and cleaned, and then the hot compression test was carried out on the Gleeble-1500D hot-compression test machine (Data Sciences International, INC., Delaware City, DE, USA). The heating rate of the hot-compression test is approximately 10 °C/s, and the compression is carried out after the temperature rises to the corresponding temperature holding time of 5 min. The equipment automatically collects parameters such as dynamic stress, dynamic strain, and temperature feedback; the compression strain is 60%, and the strain rate is 0.001 s^−1^–1 s^−1^. A high-purity Ar gas atmosphere protects the whole process of hot compression.

Scanning electron microscopy (SEM) and transmission electron microscopy (TEM) were used to detect the microstructure in different states. Backscattered electron (BSE) and electron backscattering diffraction (EBSD) microstructure observations were carried out on the FEI Quanta 200FEG field emission SEM (Thermo Fisher Scientific Inc., Waltham, MA, USA). The samples used for SEM observation were prepared by standard mechanical grinding and electrolytic polishing. The sample was electrolytically polished using 10% perchloric acid +30% butanol +60% methanol solution at −25 °C and 20 V. Experiments with selected area electron diffraction (SAED), high-resolution transmission electron microscopy (HRTEM) observation, and high-angle annular dark field (HAADF) images were performed on the FEI-TecnaiTalos F200x (Thermo Fisher Scientific Inc., Waltham, MA, USA). The foils for TEM observation were prepared by automatic twin-jet electropolishing using an electrolyte with the same composition as that used for SEM sample preparation. EBSD diagrams were processed and analyzed using the Channel 5 and Aztec 2.1 software provided by Oxford Inc. (Oxford, UK). TEM images were captured and analyzed using the TIA offline V326 software.

## 3. Results and Discussion

### 3.1. Analysis of Microstructure and Mechanical Properties of the Hot-Compressed TiAl Alloy

The composition of the Ti-46Al-8Nb-2.5V (at.%) cast alloy, which was analyzed by EDS, is Ti-45.6Al-8.2Nb-2.6V. The microstructure of the as-cast alloy is shown in Figure 1a, and the corresponding XRD pattern is shown in Figure 1b. In Figure 1a, the α_2_ phase is shown in light gray contrast, the γ phase in dark gray contrast, and the β_0_ phase in bright white contrast. The main phase is the γ phase. The middle of Figure 1a shows a remarkable parallel fine lamellar structure, which is typical of (α_2_ + γ) lamellae. This unique structure ensures that the TiAl alloy has excellent creep resistance at high temperatures [5,31].

In this experiment, the alloy was poured into the investment casting shell preheated at 800 °C and cooled under a vacuum environment, and the cooling rate was relatively slower, several times lower than that of the alloy cooled in the metal mold. Therefore, the interlamellar spacing of the TiAl alloy in Figure 1a was considerably thicker than the metal-mold solidified microstructure, and the size, morphology, and distribution of the phase were also different. There is no obvious columnar crystal feature. These differences are mainly related to the cooling rate. Yang et al. conducted in-depth research on the curve of continuous cooling transformation behavior of a high Nb-containing the TiAl alloy. With the decreased cooling rate, the phase changes entirely, and the layer thickness and spacing in the (α_2_ + γ) lamellar colonies increase significantly [32].

The stress-strain curves of the Ti-46Al-8Nb-2.5V alloy at different strain rates and temperatures are shown in Figure 2. It can be seen that the higher the temperature and the lower the strain rate, the lower the peak stress of compression. Moreover, the values of σ_p_ and $\ln\overset{\cdot }{\varepsilon }$ and the values of 1/T are strictly positively correlated. At the early stage of hot compression, the compressive stress increases sharply to the peak stress within a small strain range under the action of work-hardening, where the main factor is dislocation movement and piling up. In the (α_2_ + γ) lamellar microstructure of the TiAl alloy, the inter-spacing between lamellae is at the nanometer level, and a large number of straight α_2_/γ phase boundaries and γ/γ twin boundaries greatly shorten the distance of dislocation movement. The degree of working hardening of the alloy is intensified [33,34]. With the increase of compression strain, the accumulated energy of dislocation piling and entanglement is enough to cause the occurrence of softening mechanisms such as dynamic recrystallization (DRX), dynamic recovery (DRV), and spheroidization, ultimately leading to the gradual reduction of flow stress of the TiAl alloy, which is strain-softening. The proportion formula of strain-softening is shown in Formula (1), and its calculation results are shown in Table 1.
η = (σ_p_ − σ_ss_)/σ_p_ × 100% (1)
where σ_p_ refers to the highest stress value, while σ_ss_ corresponds to the steady-state stress, when the first and second times of θ = dσ/dε equal 0, respectively. When considering the strain-softening ratio (η), both work-hardening and strain-softening should be taken into account. It is important to note that work-hardening occurs throughout the deformation process, providing continuous energy for the DRV and DRX processes. However, the DRX process does not start at the beginning. The general definition of ε_DRX_ = 0.8εp marks the beginning of the DRX process.

Table 1 calculates the strain-softening ratio η of strain rate and temperature, respectively. Evidently, the strain-softening of the TiAl alloy is more obvious with higher temperatures, which is mainly reflected in two aspects: the influence of temperature on peak stress is mainly reflected in the enhancement of thermal activation energy, which promotes the element diffusion and dislocation movement, and dramatically promotes the DRV and DRX processes. At the same time, the increase in deformation temperature is conducive to reducing the critical shearing stress CRSS and initiating additional dislocation slip systems. In addition, it is worth pointing out that when the compression temperature of the TiAl alloy is higher than the T_eu_ (T_eu_ is typically higher than 1100 °C), the order and disorder transformation between the α and α_2_ phases will inevitably occur and promote the increase of volume fraction of the α phase and even the coarsening of the inter-lamellar spacing [9]. This microstructural evolution will severely affect the flow stress during the compression process.

Strain rate is another critical factor affecting the work-hardening and strain-softening characteristics of the TiAl alloy. The increase in deformation resistance often accompanies the increase in strain rate. Even in the strain-softening stage, it still has an intense work-hardening effect; that is, the DRXed grains still have an intense work-hardening effect and a low strain-softening proportion at a higher strain rate. When the strain rate is reduced, the deformation resistance and strain-softening effect are reduced simultaneously, providing sufficient time for dislocation slip and climb, thereby reducing the dislocation appreciation and work-hardening effect. When the strain rate is reduced to 0.001 s^−1^, the strain-softening proportion is decreased. The work-hardening effect of the alloy may be overly low at extremely low strain rates, leading to its insignificant work-hardening and strain-softening characteristics.

In contrast to single-phase alloys, the mechanism of strain rate effects on plastic deformation resistance and flow stress in multi-phase TiAl alloys is complex and may involve softening processes of lamellar structures such as rotation, bending, tilting, and local deformation. The DRX process is also different from that of a single-phase alloy, and the reconstruction analysis of this process should be combined with the strain-softening curve and microstructure. Figure 3 shows the TEM microstructure of different regions of the Ti-46Al-8Nb-2.5V alloy after hot compression at 1523 K. Figure 3a shows the grain at the early stage of work-hardening, where dislocation is essentially activated and piled up toward the α_2_/γ phase interface, providing energy accumulation for DRX and DRV. Figure 3b shows the bending of (α_2_ + γ) lamellae. Bending and tilting are the primary morphologies of lamellae during thermal deformation. Relative to the lamellae, the phase boundary of the lamellae is hard and oriented perpendicular to the compression direction, as shown in Figure 4a. The hard orientation of the lamellae makes it too difficult to accumulate strain energy to facilitate DRV and DRX processes. The lamellae with this orientation often form residual lamellae after hot compression (Figure 4a). However, lamellae that deviate from this hard orientation are more likely to tilt, bend, or even DRX under compressive stress, as shown in Figure 3b. The lamellae in the bending region have a specific deviation from the original lamellae in terms of diffraction but still retain the orientation relationship between the γ and α_2_ phases. Figure 3c,d show continuous dynamic recrystallization (CDRX) dominated by a grain-boundary arch and discontinuous dynamic recrystallization (DDRX) dominated by new-oriented grain nucleation [35].

The EBSD diagram of the central region organization of the Ti-46Al-8Nb-2.5V alloy deformed by hot compression at 1250 °C/0.1 s^−1^ is shown in Figure 4. The band contrast image of Figure 4a can be used to divide the microstructure into the DRX region and residual lamellar region according to the contrast and morphological characteristics, in which the interlamellar interface of the larger residual lamellar colony is perpendicular to the compression direction (CD). In contrast, the DRX region is composed of refined equiaxed grains with their size ranging from sub-micron to less than 10 μm. After the hot compression process, the Ti-46Al-8Nb-2.5V alloy is still composed of the α_2_ phase, β_0_ phase, and γ phase, and the orientation relationships among the three phases are calculated using the pole figures, as shown in Figure 4b. It can be seen that after hot compression, the α_2_ phase has a strong preferred orientation, forming a typical (0001) basal texture, and the pole figure intensity factor is 20.59. There are no direct orientation relationships between the γ phase and the β_0_ phases, especially since the {111}γ and {110}β_0_ plane intensity factors are low. The peak values of the α_2_ phase (red circle) and γ phase (black circle) pole figures in Figure 4b were calibrated. The γ phase and α_2_ phase in the residual lamellar colony satisfy the Blackburn orientation relationship, which is confirmed by the diffraction spot results in Figure 3b.
$$\{111\}\gamma //(0001)\alpha_{2},\;<1\bar{1}0]\gamma //<11\bar{2}0>\alpha_{2}$$

Combining the morphology observed in Figure 4a with the confirmation of the Blackburn orientation relationship in Figure 4c, it can be concluded that the α_2_ phase and γ phase primarily confirm the Blackburn orientation relationship in the residual lamellar region. In contrast, the α_2_ phase and γ phase in the DRX region do not maintain this relationship. Similar to the γ phase, the β_0_ phase does not show a significant DRX texture after hot-compression, and its pole figure intensity factor is low. After the hot-compression process, the volume fraction of the γ phase is 51.7%, the volume fraction of the α_2_ phase is 39.0%, and the volume fraction of the β_0_ phase is 9.3%.

After hot compression, the strain energy storage is different between the γ, the α_2,_ and the β_0_ phases, and the strain energy storage in different regions is also quite different [36,37]. Figure 5 analyzes these problems from multiple angles. The KAM diagram in Figure 5a can show the kernel average misorientation within the crystal and then calculate the dislocation density ρ_GND_:ρ_GND_ = 2A_KAM_/μb 
where the A_KAM_ value represents an average of the KAM for the selected region, which is derived through EBSD software. The parameter μ represents the step length chosen in the EBSD experiment, whereas b refers to the length of the Burgers vector [38]. It is important to note that the A_KAM_ value can, to some extent, reflect the dislocation density and the degree of work-hardening.

The KAM value in the residual lamellae region is significantly higher than that in the DRX region, as shown in Figure 5a. This indicates that dislocation packing in the grains in the DRX region is sufficiently released, while numerous dislocations still accumulate in the residual lamellae. To additionally analyze the intercrystalline misorientation, the α_2_ phase, γ phase, and β_0_ phase were taken into consideration, with the threshold values set as Recrystallized grain, Substructured grain, and Deformed grain, respectively. As illustrated in Figure 5b, the γ phase consists mainly of Recrystallized grain and Substructured grain, with the grains in the residual lamellar region being fundamentally Substructured grain. In contrast, the Recrystallized grains of the γ phase mostly exhibit fine equiaxed crystals and are distributed in the DRX region. Additionally, the volume fraction of the Recrystallized β_0_ phase grain reaches 82.72%, indicating that the β_0_ phase undergoes relatively sufficient DRX during the hot-compression process at 1250 °C and plays a role in coordinating alloy deformation as a high-temperature softening phase. The α_2_ phase exhibits a similar overall misorientation as the γ phase, with the large residual lamellae being essentially Substructured grain. In conclusion, high-strain energy is accumulated after hot compression in the γ phase and the α_2_ phase, especially in the residual lamellae region, while the β_0_ phase has relatively low strain energy storage.

It is essential to note that in this study, the DRX region in Figure 4a and the Recrystallized grain in Figure 5b are two distinct concepts. The DRX region refers to the microstructure morphology formed through the dynamic recrystallization process, which is differentiated from the residual lamellar region and consists of all the equiaxed grains. In contrast, the Recrystallized grain is defined based on the overall misorientation within the crystal, representing grains with lower stored strain energy after the hot-compression process.

### 3.2. Analysis of Microstructure Evolution of the TiAl Alloy after Hot-Compression and Subsequent Annealing at Service Temperature

As a high-temperature structural material with excellent properties, TiAl alloys have high microstructural stability in a range of service temperatures and undergo frequent phase transformations after thousands of hours of thermal exposure at service temperatures [16,17,18]. However, in this experiment, the 1250 °C/0.1 s^−1^ sample was annealed at 900 °C for 30 min and 60 min, respectively. The distribution of the different phases and the statistics of the phase volume fraction are shown in Figure 6. Compared with Figure 4c, the TiAl alloy in Figure 6 underwent a significant phase transformation after annealing for 30 min and 60 min, respectively. After annealing for 30 min, the volume fraction of the α_2_ phase changed from 39.0% to 4.6%, the volume fraction of the β_0_ phase decreased from 9.3% to 6.2%, and both the α_2_ phase and β_0_ phases transformed to the γ phase after annealing. The volume fraction of the γ phase increased from 51.7% to 89.2%. After 60 min of annealing, the phase volume fraction of the alloy shows little change compared to the 30 min case. The volume fraction of the α_2_ phase and β_0_ phase decreases slightly while the volume fraction of the γ phase increases slightly.

The degree of deformation in different regions of the hot-compressed sample is different, and the schematic diagram of the compression strain distribution is shown in Figure 7a. The region with a large span is selected for analysis, and its microstructure is shown in Figure 7. In the band contrast image in Figure 7a, the gradual change of the microstructure can be significantly observed. The boundaries between the DRX region and the residual lamellar region are apparent. The left region of the alloy in Figure 7a is a large deformation region, and the degree of deformation gradually decreases to the right region. In the large deformation region, more significant DRX occurs, and fewer residual lamellae are detected, decreasing gradually. Moreover, DRX is preferably activated in the non-lamellar region consisting of the β_0_ and γ phases (Figure 1a) and is accompanied by grain growth, continuous work-hardening, and new round DRX. Figure 7b shows the phase distribution and γ twin boundary (actual twin boundary) distribution after annealing at 900 °C for 30 min. In the as-cast lamellar microstructure of the TiAl alloy, a straight twin boundary is a typical microstructural feature. The bending degree of the (α_2_ + γ) lamellae can be inferred by comparing the distribution of residual lamellar in Figure 7a with the length and distribution density of the twin boundary in Figure 7b.

The volume fraction of the phase in Figure 7b is close to that in Figure 6, but the distribution of the α_2_ phase has a hierarchical distribution, and the distribution of the β_0_ phase has a significant aggregation. The volume fraction of the α_2_ phase in the DRX region is significantly higher than that in the residual lamellae region, mainly because the cumulative strain energy of the α_2_ phase in the lamellae is considerably higher than that in the DRX region, which promotes the complete transformation of the α_2_ to γ phase. The annealed β_0_ phase has a significant aggregation and is mainly distributed in the DRX region and the non-lamellar region of the compressed microstructure. While the β_0_ phase is essentially not detected in the residual lamellar region. Due to the limitation of detection accuracy of EBSD technology, no significant phase transformations of γ → β_0_ and γ → α_2_ were observed after the annealing at 900 °C. The TEM technique will be used in the subsequent analysis.

In Figure 7c, the γ phase of the sample annealed at 900 °C was analyzed by intragranular overall misorientation. It can be seen that Recrystallized grain accounts for 21.1% and is mainly distributed in the DRX region, although it is basically in the non-lamellar region. Substructured grain accounts for 62.4% and is mainly distributed in the lamellar region with slight deformation. Deformed grain, which accounts for 16.5%, is mainly located in the curved residual lamellae area with large deformation, including the DRX area. Strain-softening and work-hardening are deformed simultaneously, which results in the growth of DRX grains, repeated work-hardening, and a new around of the DRX process.

It can be seen that after annealing at 900 °C for 30 min or 60 min, the TiAl alloy underwent significant α_2_ → γ and β_0_ → γ phase transformations, and the phase transformation rate was much higher than that in the heat exposure experiment without stress [16,17,18]. The strain energy storage of the hot-compressed sample provides the driving force for the rapid phase transformation of the TiAl alloy at 900 °C. By analyzing the phase transformation during heat treatment and heat exposure processes, it can be seen that the conventional α_2_ → γ and β_0_ → γ phase transformations follow a certain orientation relationship [16,17,18,39], but whether the phase transformation process driven by high strain energy follows this orientation relationship needs to be further analyzed.

In the present study, the close-packed planes of the (0001)α_2_ plane of the D0_19_ structure and the {111}γ planes of the L1_0_ lattice structure have a very similar atomic arrangement, and the main difference between the two phases is the stacking sequence of the closely packed planes. The α/α_2_ phase of the total dislocation on the basal plane is started internally and decomposed into Shockley partial dislocation.
$$1/3<11\bar{2}0>\;\to \;1/3<1\bar{1}00>\;+\;\mathrm{CSF}\;+\;1/3<01\bar{1}0>$$

Complex stacking fault causes the stacking sequence to change from …ABABAB… to …ABCABC… (or …ACBACB…), namely the D0_19_ lattice structure transformed to the L1_2_ structure, while elemental diffusion causes the Ti atoms in the 1/2[001] position of L1_2_ to be replaced by Al atoms, and the γ phase of the L1_0_ structure is formed [6]. During the phase transformation, the α phase and γ phase strictly follow the Blackburn orientation relationship. However, the phase transformation between the α/α_2_ phase and γ phase involves elemental diffusion, which is extremely sensitive to temperature, and the element is extremely difficult to diffuse at the service temperature [40,41,42]. Therefore, in the conventional thermal exposure experiment, the α_2_ phase still has a minimal range of phase transformation after thousands of hours and does not cause long-distance diffusion or evident phase transformation.

In this experiment, the hot-compressed Ti-46Al-8Nb-2.5V alloy was annealed at 900 °C for 30 min. During this annealing process, a significant microstructure evolution occurred as most of the α_2_ phase in the alloy transformed into the γ phase. This transformation is primarily attributed to the high dislocation density, which provides numerous elemental diffusion channels for the phase transformation. The orientation relationship followed in the phase transformation process needs to be further analyzed. The α_2_ → γ phase transformation is challenging to be dynamically monitored in the current time, but the law of the phase transformation can be analyzed by calculating the texture characteristics of the α_2_ and γ phases.

The pole figure characteristics of the α_2_ phase after hot compression in Figure 4 show a typical (0001)α_2_ basal plane texture after hot compression. The (0001)α_2_ close-packed plane of most of the recrystallized grains and the residual lamellar α_2_ plates are vertical to the compression direction of the sample. The (0001)α_2_ basal texture deviating from the standard direction less than 20° is statistically analyzed, and its grain distribution is shown in Figure 8a. In Figure 8d, the grain volume fraction of the (0001)α_2_ basal texture accounts for 69.0% of the α_2_ phase and 26.9% of all phases. After analyzing the γ phase pole figure diagram (Figure 4b) and texture characteristics (Figure 8d) after hot compression, the characteristics of the γ phase texture are not prominent, {111}γ plane texture accounts for 19.7%, {110}γ plane texture accounts for 18.3%, {100}γ plane texture accounts for 6.6% (in this statistic, the c/a value of the γ phase is assumed to be 1, that is, a FCC structure is assumed). The hot-compressed γ phase has no specific outstanding texture.

After being annealed at 900 °C, if the α_2_ → γ phase transformation process follows the Blackburn orientation relationship, the α_2_ phase with {0001}α_2_ basal plane texture will be transformed completely into the γ phase with {111}γ plane texture. In this case, the {111}γ plane texture of the annealed alloy will increase significantly. The volume fractions of the {111}γ plane texture assumed based on the Blackburn orientation relationship and the actual {111}γ plane texture of the α_2_ → γ phase transformation are compared, as shown in the data table in Figure 8d. The assumed value of {111}γ plane texture is considerably higher than the actual value. Moreover, after heat treatment, the {111}γ plane texture ratio of the γ phase slightly decreased, while the {110}γ texture gradually increased. This fully demonstrates that when the high-strain energy storage Ti-46Al-8Nb-2.5V alloy is annealed at 900 °C, the α_2_ → γ phase transformation process does not follow the Blackburn orientation relationship. After annealing, the alloy’s texture characteristics are significantly weakened due to the decrease in the volume fraction of the α_2_ phase.

Because the sample was quenched after hot compression, the phase composition of the alloy deviated from the equilibrium state to a certain extent, especially the α_2_ phase solid dissolved the supersaturated Al element, forming a large number of anti-vacancy defects, which also provided diffusion channels for the service temperature phase transformation of the alloy. However, in the published studies, the α_2_ → γ phase transformation caused by the anti-vacancy diffusion follows the Blackburn orientation relationship. In this study, the defects of a large number of dislocations also provide driving forces for the diffusion and phase transformation of the alloy. However, Figure 9 shows the KAM statistics of the alloy after annealing at 900 °C. The A_KAM_ value of the alloy is 0.5632 after 30 min of annealing and 0.5792 after 60 min of annealing. Although there is a significant α_2_ → γ phase transformation, compared with Figure 5a, it is found that the dislocation density of the annealed alloy not only does not decrease but increases to a certain extent.

By analyzing the change in A_KAM_ value, it can be explained from two aspects. First, as the dislocation density difference between the DRX region and the residual lamellae region is too large in Figure 9, the A_KAM_ value is more affected by the volume fraction of the residual lamellae in the particular region. The second is the difference in crystal structure between the α_2_ phase and γ phase. The c/a value of the γ phase of the L1_0_ structure is very close to the FCC structure, with four groups of close-packed {111} plane. The γ phase can be activated by all dislocations b = 1/2<110], b = <001], and b = 1/2<11$\bar{2}$]. These all dislocations preferentially slip on the {111}γ plane. It is also possible to achieve the possibility of slipping on other crystal planes through cross-sliding. However, the Shockley partial dislocation in the γ phase can achieve γ→α_2_ phase transformation, and the Frank partial dislocation can lead to the formation of γ phase twins and pseudo-twins. The α_2_ phase with D0_19_ structure has only one set of (0001) close-packed planes, and the deformation of the material is basically completed by the <a> type total dislocation b = 1/3<11$\bar{2}$0] and <2a + c> type total dislocation. The number of sliding systems that can be activated is much lower than the γ phase, and the solid solubility of α_2_ relative to O, C, N, and other interstitial elements is much higher than the γ phase and β_0_ phase. The interstitial elements will significantly improve the activation energy of dislocation motion. The number of dislocations that can be activated was significantly higher in the γ phase than in the α phase, which may also be the reason why the average dislocation density of the alloy does not decrease but slightly increases after the α_2_ → γ phase transformation occurs in annealing.

Figure 10 shows the TEM microstructure of the Ti-46Al-8Nb-2.5V alloy after hot compression after annealing at 900 °C for 60 min with DRX γ phase. The γ phase in Figure 10a is the matrix phase, and the dislocation density in the γ phase in the image is low, but the α_2_ phase with nanometer or even subnanometer size is precipitated. That is, the γ → α_2_ phase transformation occurs. The newly precipitated α_2_ phases are marked as ①–④. According to the morphological analysis of the phase transformation, the α_2_ phase precipitated in the γ phase follows the Blackburn orientation relationship [43]. The broad interface of the two phases is very straight, and the habit plane is (0001)α_2_//{111}γ. It can be clearly seen that in Figure 10a, the broad interfaces of ①-α_2_ and ②-α_2_ are parallel to each other, while ③-α_2_ and ④-α_2_ are at a certain included angle, which is mainly with γ phase, where there are four groups of interlacing close-packed planes, different α_2_ phases nucleate and grow on different close-packed planes of γ phase, forming a variety of growth directions of α_2_ phase. Although the alternating movement of the Shockley partial dislocation can achieve α_2_ → γ and γ → α_2_ phase transformation, these two phase transformations do not have the reversibility of phase transformation morphology. The main reason is that the number of close-packed planes of the α_2_ phase and γ phase is different. By exploiting the irreversibility of the phase transformations between the phases, the alternating and repeated occurrence of the two-phase transformation can be achieved by a cyclic heat treatment with multiple heating and cooling, which can refine the microstructure and improve the mechanical properties of the TiAl alloy [44,45].

Figure 10b–d show that a new precipitated Ti_2_Al phase (a = b = 3.040, c = 1.369 nm) is formed on the broad interface between the γ and α_2_ phases. This phase has a length of about 300–400 nm and a width of about 80 nm. The Ti_2_Al phase has the same crystal structure and extremely similar crystal parameters as the Ti_2_AlN phase (P63/mmc, a = b = 2.985, c = 13.567) and the Ti_2_AlC phase (P63/mmc, a = b = 3.040, c = 13.600) [46,47,48]. He et al. determined that the close-packed planes of the Ti_2_Al phase are the same as the α_2_ phase and γ phase, the order of the close-packed planes is AB’CBC’BA, and the stacking period is three times that of α_2_ and twice that of γ phase and inferred that the Ti_2_Al phase is the transient phase between the α_2_ and γ phase. The orientation relationship between the Ti_2_Al phase, γ phase, and α_2_ phase is calibrated in Figure 10c [49]:$$[11\bar{2}0]\alpha_{2}//[011]\gamma //[11\bar{2}0]{\mathrm{Ti}}_{2}\mathrm{Al},$$
$$(0001)\alpha_{2}//(11\bar{1})\gamma //(0001){\mathrm{Ti}}_{2}\mathrm{Al},\;(01\bar{1}0){\mathrm{Ti}}_{2}\mathrm{Al}//(01\bar{1}0)\alpha_{2}$$

The calibration results in Figure 10c show that the γ1 phase and γ2 phase are in the orientation relationship of the twin, but the twin interface of the (111)γ1/γ2 is not parallel to the habit plane of the (0001)α_2_ in Figure 10c, but there is an angle of approximately 69.55°. This experimental phenomenon is due to the nucleation of the α_2_ phase on the (11$\bar{1}$)γ1 close-packed plane but the nucleation of the γ2 twin on the (111)γ1 close-packed plane. Since there is no obvious orientation relationship between the γ2 phase and the α_2_ phase, the interface between the two phases in Figure 10b is characterized by a curved interface with close-packed plane dislocation distance rather than a straight one with periodicity dislocation.

However, in previous studies, the Ti_2_Al phase of the 6H structure was found at the ledge plane position of the α_2_ phase and the γ phase or precipitated in the γ phase alone [21]. In the creep test on the service temperature of the TiAl alloy, the volume fraction of the α_2_ phase decreases with the creep test, and then the secondary phase precipitates in the γ phase discontinuously dispersed. This is because the solid solubility of the α_2_ relative to H, C, N, O, and other interstitial elements is much higher than that of the γ phase [50,51,52,53], and the volume fraction of the α_2_ phase significantly decreases during the service temperature annealing of both the creep test and the present experiment. Due to the low solubility of γ relative to the interstitial elements, the interstitial elements will be precipitated as compounds in the γ phase. The Ti_2_AlC or Ti_2_AlN compounds mentioned above are also more likely to precipitate in the γ phase. Since determining interstitial elements is very difficult, more professional techniques are needed to determine the proportion of interstitial elements in the precipitated phase and whether it is the Ti_2_Al or other secondary phases.

## 4. Conclusions

In the present study, the high Nb-containing Ti-46Al-8Nb-2.5V alloy was subjected to hot-compression experiments and subsequent annealing at 900 °C. Significant microstructural evolutions occur during both hot compression and annealing, and the conclusions of the study are as follows.
The stress-strain curves of Ti-46Al-8Nb-2.5V alloys during hot-compression show typical work-hardening and strain-softening features, with the peak compressive stress positively correlated with the strain rate and negatively correlated with the compression temperature. The alloy undergoes significant DRX during hot-compression, with elevated values of the strain-softening rate and a large number of fine equiaxed crystals in the microstructure;After hot-compression, the microstructure consists of an equiaxed grain region and a residual lamellar colony region consisting of α_2_ and γ phases with high internal strain. The volume fractions of the α_2_ phase, the γ phase, and the β_0_ phase are 51.7%, 39.0%, and 9.3%, respectively;When the hot-compression samples are annealed at 900 °C for 30 min, evident phase transformations occur, and the volume fraction of the α_2_ phase and the β_0_ phase decrease to 4.6% and 6.2%, respectively. The volume fraction of the γ phase increased to 89.2%. When the annealing time is extended to 60 min, the microstructure morphology and the volume fraction of the phases did not change significantly. The compressed samples annealed at 900 °C/30 min reached a relatively stable microstructure;After hot compression, the α_2_ phase exhibits a strong (0001)α_2_ basal plane texture, with a pole figure strength factor of 20.59. The grains with (0001)α_2_ texture account for 69% of the volume fraction of the α_2_ phase and 26.9% of all phases. After annealing at 900 °C, a significant α_2_ → γ phase transformation occurred, but the volume fraction of {111}γ texture did not increase, and the α_2_ → γ phase transformation did not wholly follow the Blackburn orientation relationship. Additionally, the γ → α_2_ phase transformation also occurred within the γ phase during the annealing process but followed the Blackburn orientation relationship.

## Figures and Tables

**Figure 1 materials-16-06176-f001:**
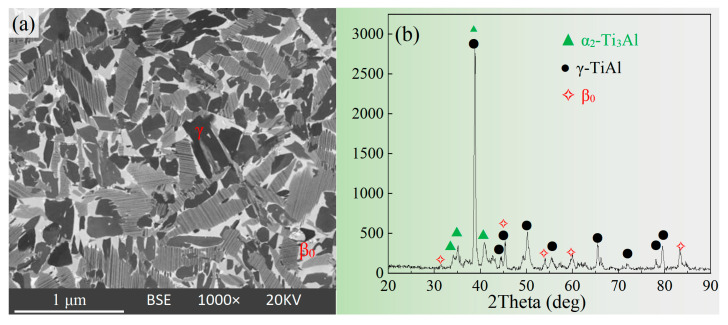
BSE and XRD pattern of the Ti-46Al-8Nb-2.5V alloy. (**a**) BSE microstructure; (**b**) XRD pattern.

**Figure 2 materials-16-06176-f002:**
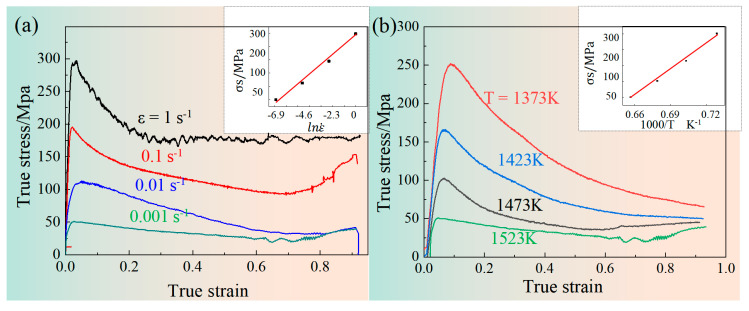
Stress-strain curves of the Ti-46Al-8Nb-2.5V alloy subjected to Gleeble hot compression at different strain rates and temperatures: (**a**) different strain rates at 1250 °C (1523 K); (**b**) different temperatures with ε is 0.1 s^−1^.

**Figure 3 materials-16-06176-f003:**
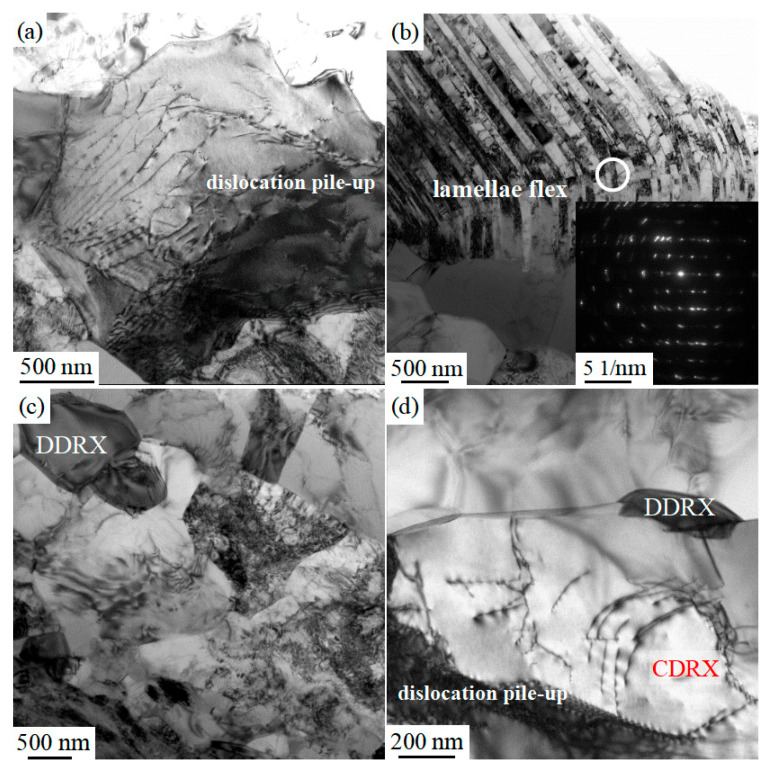
Images of TEM microstructure of the Ti-46Al-8Nb-2.5V alloy after hot compression tested at 1250 °C: (**a**) dislocation pile-up; (**b**) microstructure of the tilted lamellar microstructure; (**c**,**d**) dislocation pile-up and DRX region.

**Figure 4 materials-16-06176-f004:**
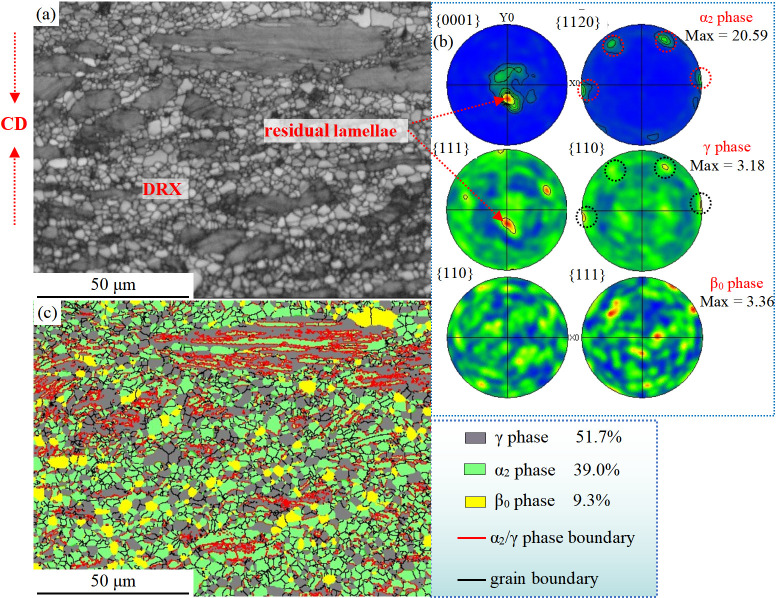
EBSD microstructure of the Ti-46Al-8Nb-2.5V alloy after 1250 °C/0.1 s^−1^ hot compression: (**a**) band contrast diagram; (**b**,**c**) pole figures; (**d**) phase map and certain grain boundary.

**Figure 5 materials-16-06176-f005:**
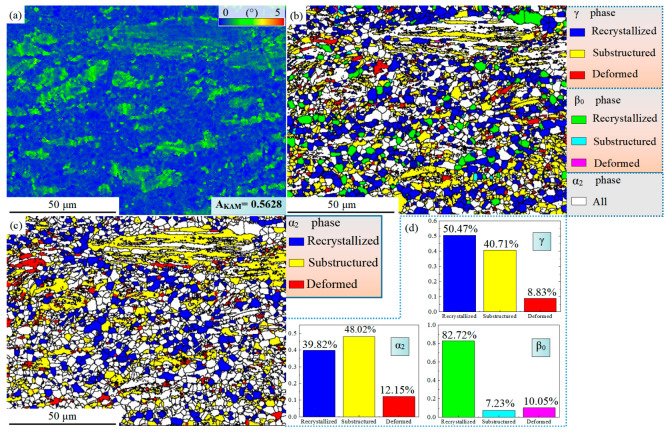
EBSD images of the Ti-46Al-8Nb-2.5V alloy after 1250 °C/0.1 s^−1^ hot compression: (**a**) KAM image; (**b**) overall crystal misorientation of the γ and β_0_ phases; (**c**) overall crystal misorientation of the α_2_ phase; (**d**) statistics of the Recrystallized grain, Substructured grain, and Deformed grain of all phases.

**Figure 6 materials-16-06176-f006:**
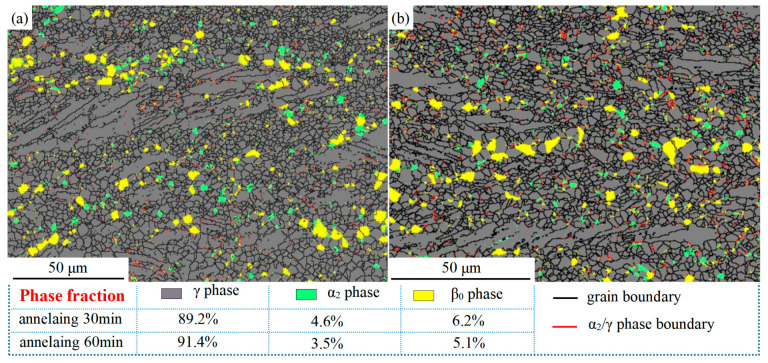
EBSD images of the distributions and volume fractions of all phases of the Ti-46Al-8Nb-2.5V alloy after 1250 °C/0.1 s^−1^ hot compression and subsequent annealing at 900 °C: (**a**) annealed for 30 min; (**b**) annealed for 60 min.

**Figure 7 materials-16-06176-f007:**
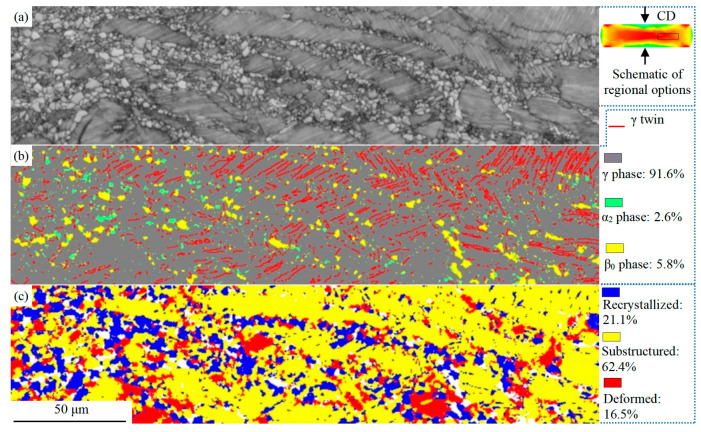
EBSD images of the Ti-46Al-8Nb-2.5V alloy after 1250 °C/0.1 s^−1^ hot compression and subsequent annealing at 900 °C: (**a**) band contrast diagram; (**b**) distributions and volume fractions of all phases; (**c**) overall crystal misorientation of the γ phase.

**Figure 8 materials-16-06176-f008:**
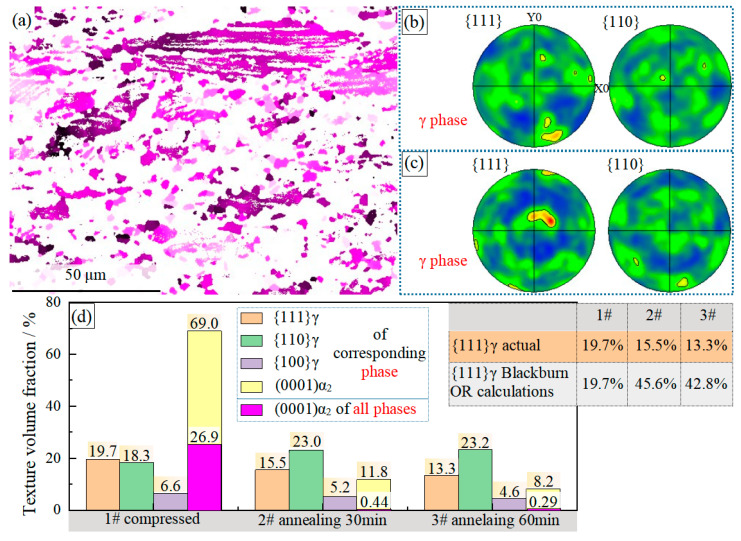
EBSD images of the Ti-46Al-8Nb-2.5V alloy after different processes: (**a**) distribution of the (0001)α_2_ basal plane texture; (**b**) pole figures of the γ phase after hot compression and annealing for 30 min; (**c**) pole figures of the γ phase after hot compression and annealing for 60 min; (**d**) statistics of the volume fractions of the textures.

**Figure 9 materials-16-06176-f009:**
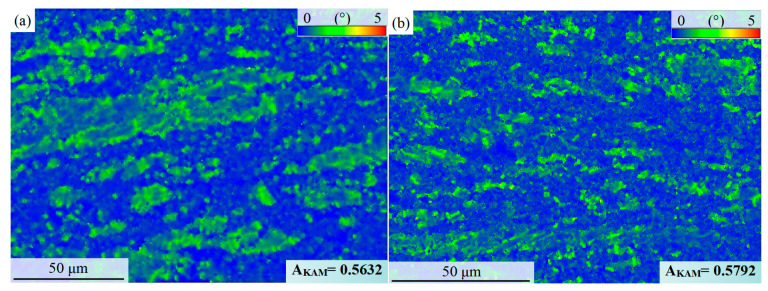
KAM images of the Ti-46Al-8Nb-2.5V alloy after 1250 °C/0.1 s^−1^ hot compression and subsequent annealing at 900 °C: (**a**) annealed for 30 min; (**b**) annealed for 60 min.

**Figure 10 materials-16-06176-f010:**
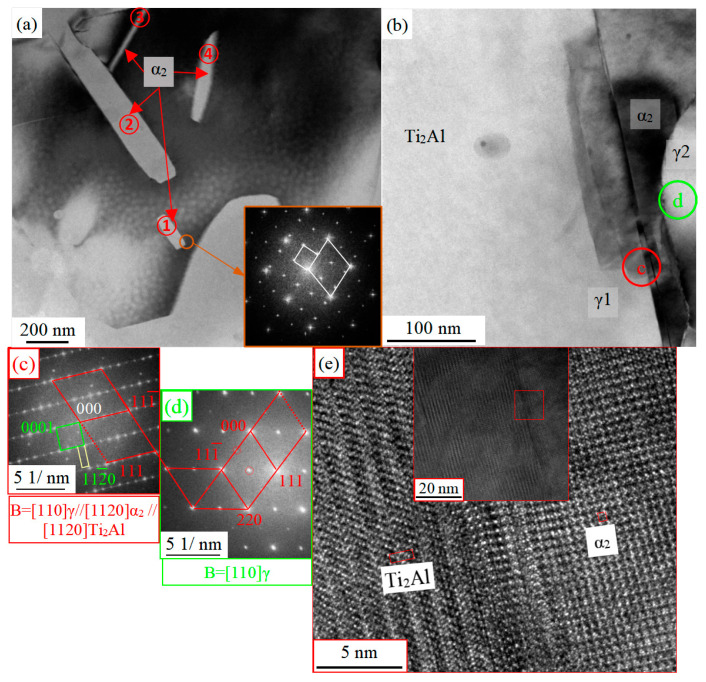
Images of TEM microstructure of the Ti-46Al-8Nb-2.5V alloy after 1250 °C/0.1 s^−1^ hot compression and subsequent annealing at 900 °C for 60 min: (**a**,**b**) the microstructure internal the gamma phase; (**c**) SEAD patterns corresponding to the red circle region in (**b**); (**d**) SEAD patterns corresponding to the green circle region in (**b**); (**e**) HRTEM images corresponding to the red box region.

**Table 1 materials-16-06176-t001:** Strain-softening value of the Ti-46Al-8Nb-2.5V alloy subjected to Gleeble hot compression at different strain rates and temperatures.

Strain Rate	η Value	Temperature	η Value
1 s^−1^	41.47%	1523 K	73.77%
0.1 s^−1^	52.35%	1473 K	69.78%
0.01 s^−1^	71.96%	1423 K	64.91%
0.001 s^−1^	59.46%	1373 K	59.46%

## Data Availability

The data that support the findings of this study are available from the first author, /Shouzhen Cao/, upon reasonable request.

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
