# Peer review of "Microstructure Evolution of the Ti-46Al-8Nb-2.5V Alloy during Hot Compression and Subsequent Annealing at 900 °C"

_materials, 2023, doi:10.3390/ma16186176_

Round 1

Reviewer 1 Report

The manuscript is well written, kindly accept in the present form. I Recommend the manuscript can be accepted for publication.

Author Response

Thank you very much for reviewing and recognizing the content of the manuscript. Your recognition greatly encourages the authors. Once again, we would like to express our heartfelt gratitude to you.

Reviewer 2 Report

Microstructure evolution of Ti-46Al-8Nb-2.5V alloy during high temperature compression and subsequent annealing at  service temperature is very interesting paper! Minor improvement is required!

Line 3: subsequent annealing at  service temperature (Please to add a value of this temperature)

Line 24: Multiple crystal-oriented α2 phases with nanoscale size precipitate (in which range is nanoscale precipitate)

Line 44: considerably lower than Teu. (What is exactly an eutectic temperature Teu)

Line 91: the strain rate is 0.001s-1 -1s-1 . (please to check this unit for strain rate? Strain rate has dimension of inverse time and SI units of inverse second, s−1)

Line 210; EBSD diagram (what is the full meaning of the EBSD-diagram?)

Line 333: It can be seen that after annealing at 900°C for a short time (what is a value of short time (1-2s?)

Line 337: phase transformation of TiAl alloy at low temperatures (what is an interval of low temperature)

Line 377: In the Figure , (please to put a number of Figure? 4b?)

Line 494: SEAD patterns (what is full meaning of SEAD?)

General Questions:

High-Nb containing Ti-46Al- 13 8Nb-2.5V alloy was subjected to hot compression and subsequent annealing at 900°C. What is role of Nb and V in a structure of Ti-46 Al-13.8Nb-2.5V

Line 81: Ti-46Al-8Nb-2.5V alloy was smelted by vacuum induction in a water-cooled copper crucible. Why not in a carbon crucible?

Author Response

Thank you very much for taking the time to review this manuscript (materials-2595839). The comments and suggestions are very useful for improving this paper. Regarding grammatical and expressive errors, we made direct changes in the manuscript after carefully reviewing it. The detailed responses and revisions are listed in the attached Word files.

Reviewer 3 Report

The paper entitled „Microstructure evolution of Ti-46Al-8Nb-2.5V alloy during high-temperature compression and subsequent annealing at service temperature” focuses on the microstructure instability and phase transformation of TiAl alloy with high-stress stored energy at service temperature. The tests provided are versatile, up-to-date and complimentary and the results are well presented and give a good opportunity for further investigations. I would like to recommend its publication in this journal after addressing the following recommendations:

1)     The aim of the study should be better formulated;

2)     Usually, the composition of the alloy is given in wt% rather than in at%;

3)     The caption of Table 1 should be improved.

4)     In the whole manuscript, too many abbreviations should be clarified to improve the readability of the text.

5)     The software used for the visualization of experimental results should be mentioned in section 2;

None

Author Response

Thank you very much for taking the time to review this manuscript (materials-2595839). The comments and suggestions are very useful for improving this paper. Regarding grammatical and expressive errors, we made direct changes in the manuscript after carefully reviewing it. The detailed responses and revisions are listed in the attached Word file.
